# Phase II Clinical Study on Low-Intensity-Noise Tinnitus Suppression (LINTS) for Tinnitus Treatment

**DOI:** 10.3390/brainsci15111222

**Published:** 2025-11-13

**Authors:** Konstantin Tziridis, Lara Heep, Nathalie Piwonski, Katharina Nguyen, Nikola Kölbl, Achim Schilling, Holger Schulze

**Affiliations:** 1ENT Clinic Head and Neck Surgery, Experimental Otolaryngology, University Hospital Erlangen, Waldstrasse 1, 91054 Erlangen, Germany; konstantin.tziridis@uk-erlangen.de (K.T.); nikola.koelbl@uk-erlangen.de (N.K.); 2Center for Neuromodulation and Neuroprosthetics, University Hospital Mannheim, University Heidelberg, Theodor-Kutzer-Ufer 1-3, 68167 Mannheim, Germany; achim.schilling@medma.uni-heidelberg.de; 3BG Clinic Ludwigshafen, Ludwig-Guttmann-Straße 13, 67071 Ludwigshafen, Germany

**Keywords:** hearing aids, placebo-controlled double-blinded study, tinnitus questionnaires, tinnitus distress

## Abstract

**Background/Objectives**: Tinnitus treatment is often based on coping strategies, as, still, no causal treatment is available. Based on our “Erlangen model” of tinnitus development, we treated tinnitus patients with individualized, non-masking low-intensity noise (LIN) to reduce the tinnitus loudness. **Methods**: A total of 72 adult patients with a tinnitus severity index below IV (tinnitus health questionnaire, THQ), a hearing loss not exceeding 40 dB up to 6 kHz, and without experience in hearing aid (HA) usage were included in this study. Their audiograms, tinnitus parameters, THQ scores, and optimal LIN were determined individually. Patients were double-blinded, assigned to a treatment-only (TO) or placebo-and-treatment (PT) group. The TO patients received treatment for four weeks while the PT patients initially received a placebo (low-intensity white noise) stimulation for two weeks and, hereinafter, treatment for four weeks. Every two weeks, the measurements listed above were repeated. The long-term effects on THQ were assessed four weeks after the end of treatment. The data were analyzed by parametric and nonparametric statistics. **Results**: We identified a significant decrease in the THQ score already after two (η^2^ = 0.14) and four weeks of treatment (η^2^ = 0.43), that was still present four weeks after the end of treatment (η^2^ = 0.15) in the TO group. Subjective individual patient reports hint at a possible complete suppression of the percept by LINTS. The PT group profited much less, as the initial placebo treatment seemed to counteract the LIN effects. **Conclusions**: Individually fitted LIN treatment seems to be able to suppress tinnitus, in some cases completely. The optimal fitting of the LIN is crucial for treatment success.

## 1. Introduction

Subjective tinnitus—the perception of sounds without a physical sound source—is thought to be caused by hearing loss (HL), which can be so slight that it is clinically irrelevant or even undetectable with standard audiometry [1,2,3,4]. In many cases, especially when HL is low or moderate, the phantom percept itself is perceived as more serious than HL and can lead to physical and psychological comorbidities, which, in the worst case, can drive patients to suicide [5,6]. The socioeconomic costs caused by tinnitus and the resulting comorbidities are estimated at almost €22 billion per year in Germany alone, and the number of sick days is more than twice as high as that of the average German employee [7]. Comparable figures are also available for other Western industrialized nations [8].

In clinical practice, the effective treatment of tinnitus is typically limited to counseling and teaching coping strategies within the framework of cognitive behavioral therapy (CBT; cf. [9]), as there is still an incomplete understanding of the neurophysiological mechanisms involved in the development and chronification of tinnitus. Incidentally, this is also the reason why most therapeutic approaches tested to date beyond CBT were based not on theoretical but merely on empirical considerations, with mostly modest, non-reproducible success and often without any placebo control [10,11,12,13,14,15,16]. Since tinnitus is a maladaptation of the auditory system to HL, it is not surprising that the most successful treatments for tinnitus are various methods for hearing restoration [10,17], but usually these are not available for patients with mild HL. In this placebo-controlled phase II clinical study, we tested a new, theory-driven therapeutic approach to treating tinnitus, which is based on a neurophysiological model of tinnitus development and explicitly aims to help patients with mild to moderate HL.

Over the past decade, we have developed a model for the development of tinnitus that considers the phantom percept to be a side effect of a neurophysiological mechanism that permanently optimizes information transmission into the auditory system by means of stochastic resonance (SR) in the dorsal cochlear nucleus (DCN), even in the healthy auditory system [18,19,20]. Like most models, ours also assumes that HL initially causes the development of a tinnitus percept. In fact, tinnitus only occurs when HL is caused by the reduced afferent innervation of the inner hair cells of the cochlea [21], which, as mentioned above, may be so slight that no increase in hearing thresholds can be measured in the pure tone audiogram (cf. hidden hearing loss [22]). Such a reduced input from the cochlea to the DCN can already lead to subthreshold responses of the target neurons in the DCN and, thus, the auditory input would no longer be passed on to the subsequent auditory pathway [23]. Our Erlangen model of tinnitus development [24] now assumes that SR occurs at the level of DCN neurons (fusiform cells) to enable the transmission of weak signals to the auditory pathway and, thus, improve the overall hearing ability. To this end, noise in the form of neural activity (originating from the somatosensory system [25,26]) is added to the weak input signal from the cochlea via parallel fibers also projecting to the fusiform cells. The amount of noise added is controlled by a feedback loop that constantly measures the information content of the DCN output by calculating and maximizing the autocorrelation of that output [19]. The sum of the input signal and noise is then large enough to activate the DCN neurons above the threshold, so that the auditory information, even though it is noisy, will be transmitted. According to this theory, the internally added neural noise improves overall hearing, but may also be transmitted to the auditory cortex itself, where it forms the basis for the perception of tinnitus.

Based on this mechanistic explanation of tinnitus development, we have proposed a novel treatment strategy, “Low-Intensity-Noise Tinnitus Suppression” (LINTS [24]), which has already proven promising in two proof-of-concept studies [27,28]. The basic idea of this approach is to replace the internally generated neuronal noise that is perceived as tinnitus with externally generated low-intensity acoustic noise matched to the patient’s tinnitus frequency and hearing threshold.

As we have demonstrated in our proof-of-concept studies, this acoustic low-intensity noise (LIN) is most effective in suppressing tinnitus when a narrowband noise is used that is centered on the individually perceived tinnitus pitch and has an amplitude around the individual hearing threshold. Note that this amplitude is too low to mask the tinnitus percept, so it should not be consciously perceived in everyday life when used constantly.

Based on the results of the proof-of-concept studies mentioned above, the limitations of our method are initially that mild hearing loss—at least without amplification by a hearing aid—and a tinnitus frequency within the frequency range of the devices used are required. The first limitation, in particular, mild hearing impairment, opens up treatment options for patients who do not usually receive hearing aids [29]. Our study should, therefore, be seen as a first step toward treating tinnitus with the LINTS approach, which needs to be improved and expanded in future studies.

Here, we investigate the effects of the continuous use of the LINTS approach in a placebo-controlled phase II clinical study by presenting the LIN via hearing aids without activated amplification over a time span of four weeks. We demonstrate that LINTS can reduce tinnitus-related distress, as assessed by standard questionnaires, acutely and over a longer period of time compared to the placebo treatment. The extent to which tinnitus can be suppressed seems to depend on how well the LIN can be adjusted to the patient’s individual tinnitus frequency and hearing threshold. Ideally, LIN was able to subjectively suppress the tinnitus percept completely in individual cases.

## 2. Materials and Methods

### 2.1. Subjects and Ethical Statement

Eighty-four adult tinnitus patients (median age (lower, upper quartile): 55 a (40 a, 60 a); 25 ♀) were included in this study, which was approved by the ethics committee of the Friedrich-Alexander-University Erlangen-Nürnberg (application number 35_21 B) in accordance with the Declaration of Helsinki. All patients were specifically recruited for this study, and gave their informed consent for data evaluation. Each patient received €150 as compensation after the study was completed. Inclusion criteria were as follows: tonal or narrowband tinnitus perception on at least one side and a sufficient understanding of the German language to understand the instructions and fill out the German versions of the mini-TQ12 [30], Tinnitus Health Questionnaire (THQ) [31], and Tinnitus Sample Case History Questionnaire (TSCHQ) [32]. The rationale for using the THQ was its well-described sub-scores and the very high correlation with the results of the mini-TQ12, meaning that no patient with excessively severe tinnitus was included in the study. A drawback of the THQ is that the questionnaire is geared toward long-term assessment. To overcome this, we instructed the patients to focus their responses solely on the period covered by the investigation (cf. below). Exclusion criteria were as follows: previous or current wearing of hearing aids, mini-TQ12 severity index above 3, hearing loss higher than 40 dB below 8 kHz [27,28], or a tinnitus frequency above 10 kHz. Although we have made every effort to control for various patient characteristics and study design factors, this study should be considered exploratory.

### 2.2. Audiometry and Timeline of Experiments

The timeline for the experiments is depicted in Figure 1A,B. The patients meeting the inclusion criteria were pseudorandomly separated into two groups, the LIN-**t**reatment-**o**nly (TO, n = 54) and the **p**lacebo-plus-LIN-**t**reatment (PT, n = 30) group. Grouping was balanced for gender and age. The minimal group size required by power analysis (G*Power 3.1.9.2) was 23 patients per group (critical z = 1.64, actual power 0.71). TO patients were treated immediately after HA-fitting with the individually fitted LIN while PT patients received a placebo white noise stimulation for 2 weeks before switching to the correct LIN stimulation for another 4 weeks (cf. below). The reason for this approach was that we wanted to investigate the possible long-term effects after the end of LINTS therapy without any interference. Therefore, we refrained from a “balanced” approach with the TO group receiving a WN stimulation after the end of the treatment. Furthermore, we wanted to control for “any” stimulation effect with the WN stimulation in the first two weeks in the PT group. Grouping was double-blinded, completely for the patients and up to the end of the third measurement for the investigator, where it became necessary to know if another session was needed (in the case of PT). Therefore, timeline of the experiments after initial check of inclusion and exclusion criteria was different for both groups, as the stimulation was 4 weeks for the TO and 6 weeks for the PT groups. Patients were informed prior to the study that the treatment could last 4 or 6 weeks; they did not know each other or met during the study, so no crosstalk about differences in treatment duration or other factors could be communicated.

Before treatment start, patients filled out the THQ and TSCHQ. Air conductance hearing thresholds (measured as HL in dB) as well as tinnitus frequency (TF) and tinnitus loudness (TL)—both tinnitus parameters calculated relative to HL in dB sensational level (dB SL)—were obtained following the ISO 8253-2 [33] procedures (measurement device: AT900 type 1093, measurement software: AT900 5.1.0.15; both: Mack Medizintechnik, Pfaffenhofen, Germany; headphones: DT48, Beyerdynamic, Heilbronn, Germany; or HDA200, Sennheiser Electronic GmbH & Co. KG, Wedemark, Germany). The optimal LIN was determined as described in [28]. Briefly, around the frequency of the tinnitus percept (±1 oct, 0.5 oct steps), narrowband noise with intensities ranging from −4 to 6 dB SL were presented for 40 s each. After each stimulus, the patient was asked what effect the LIN had on the percept. The optimal LIN was the one with the strongest tinnitus-soothing effect. Data were transferred to Signia Pure 312 X © hearing aids (WSAudiology—Sivantos GmbH, Erlangen, Germany) via the Connexx-software (Connexx 9.7.0.144, WSAUD A/S, Lynge, Denmark). Hearing aids were only applied to the tinnitus-affected ears (50 patients binaurally supplied; 36 TO and 14 PT); the good fitting of the HA was secured by individualized cable length and size of the earpiece. The amplification function of the hearing aids (HAs) was not activated; the device was only used for presentation of the LIN. The LIN was either presented via the internal noise generator of the device (N = 36) or via a Bluetooth-coupled iPod (Apple, CA, USA), from which the patient could only play the right LIN at the right intensity (N = 48) with a narrowband (NB) noise of one octave width. The latter approach was used, as the internal noise generator of the HAs turned out to be sub-optimal for the task, as the bandwidth of the generated noise was wider than the one octave determined to be optimal for LINTS [27,28]. Therefore, the stimulation used in these patients was sub-optimal (wideband noise, WB). Furthermore, with the iPod, we were able to test additional modifications for the LIN-stimulation like AM-modulated NB noise (N = 16, AM-frequency 40 Hz, modulation depth 20%). The idea here was to increase information content of the signal and, by that, increase autocorrelation at the level of the DCN, which might be particularly beneficial for patients with higher-frequency tinnitus, where phase locking to the unmodulated noise and, therefore, autocorrelation within the DCN should be low [34].

The appropriate stimulation for either TO or PT patients was generated by computer program developed in-house (Python, Version 3.13) for each measurement session. For the TO patients, the LIN was immediately turned on and minimally adjusted in loudness if necessary. For PT patients, without their knowledge, the LIN was not transferred to the hearing aids over the first two weeks but patients were stimulated with white noise at an intensity set to that of the hearing threshold at 2 kHz. During their next visit, the individually customized LIN was uploaded and used for the next four weeks. All patients were instructed to wear the HAs as long as possible, but at least on five out of seven days a week for at least four hours per day. The wearing duration was logged in the device and results of invalid periods, i.e., average wearing times during the two-week period below 4 h/d, were discarded (in 3 patients at one time each). For the follow-up procedures, refer to the Figure 1B. For the intra-treatment THQ score determination, the patients were instructed to focus their answers on the last four weeks. Four weeks after the end of treatment, possible long-term effects were investigated with a final THQ form. If requested after that timepoint, we provided the patients with their LIN to be used with their own in-ear devices. Of the 84 patients, 72 (86%) completed all measurements and filled out all questionnaires, only the data of those patients are used for further analyses. Of the 12 dropouts, 6 were from the PT group and all of them stopped the trial after the initial placebo phase, as they stated they had no benefit from the (placebo) LIN. The other 6 dropouts of the TO group had different reasons (medical or technical) to abort the study, but no TO patient dropped out because of missing benefit of the stimulation.

### 2.3. Statistical Evaluation

All statistical analyses were performed with Statistica 14 (TIBCO software, Palo Alto, CA, USA). Analyses were performed only on the data of the 72 patients who completed the study; dropouts were not included in the analyses. The initial HL of the patients without HA of both groups was compared using a 2-factorial ANOVA with the factors *group* and *frequency*. For detailed pairwise analyses of all datapoints, Tukey post hoc tests with correction for multiple comparisons were used. The TF and TL changes over time within the two patient groups were assessed with nonparametric statistics using Wilcoxon tests or Friedman–ANOVAs and comparisons between TO and PT groups by Mann–Whitney U-tests. As for the tinnitus parameters, nonparametric analyses were used for the assessment of the change in the THQ scores and sub-scores relative to pre-treatment, as these values were discrete as well and Shapiro–Wilk tests rejected normal distribution assumption with *p* < 0.001. Here, the change in the complete questionnaire score and the change within the different sub-score categories were investigated. Bonferroni corrections of the *p*-values were used, where needed. The effect size of the nonparametric tests is either given as Kendall coefficient of concordance (Kendall’s W) [35,36] or as η^2^ value (Pearson’s correlation coefficient squared) [37]. The interpretation of both effect size values has different limits: for Kendal’s W, values below 0.2 indicate a weak effect, between 0.2 and 0.4 a fair effect, between 0.4 and 0.6 a moderate effect, between 0.6 and 0.8 a strong effect, and above 0.8 a very strong effect. For η^2^, values between 0.02 and 0.13 indicate a weak effect, between 0.13 and 0.26 a moderate effect, and above 0.26 a strong effect. For clarification, we added the interpretation to the effect size values at the respective passage in the Section 3. The complete THQ score has a range of 0 to 84, the different sub-scores have ranges from 0 to 6 (somatic complains), 0 to 8 (sleep disturbances), 0 to 14 (auditory perceptual difficulties), and 0 to 16 (intrusiveness of tinnitus), and the combined cognitive and emotional distress with a range of 0 to 40. Note that we did not include patients with the highest severity index with more than 59 TQH score points. Therefore, the median score (interquartile range) of our patients before the start of the treatment was just 26.5 (16, 35.5); for details, see Section 3.2. The questionnaire outcome was also correlated with the patients’ age, family history of tinnitus appearance, and tinnitus duration, which were extracted from the TSCHQ.

## 3. Results

### 3.1. Study Design

Details of the study design are given in the Section 2. Briefly, of the 84 patients, 72 (86%) finished the study; of those, 48 belonged to the TO and 24 to the PT group. As mentioned in the Methods section, the dropout reason in the PT group was purely the white noise treatment (6/30, 20%), while patients aborting the study in the TO group had other reasons (6/54, 11%). All patients were inexperienced in HA use and fulfilled all inclusion and exclusion criteria. The amplification of the HA was turned off, so it was only activated to perceive the auditory environment undisturbed and for LINTS stimulation.

### 3.2. Database

The 72 patients who finished the study were separated into two groups (TO and PT). The median duration (interquartile range) of the tinnitus percept (obtained from the TSCHQ) was 14.2 a (3.6 a, 26.7 a). The wearing duration of the HA was high and not significantly different in both groups, with 7.2 h/d ± 0.9 h/d in the TO and 6.6 h/d ± 1.1 h/d in the PT patient group (*t*-test, *p* = 0.53). The distributions of gender (TO: 32 male, 16 female; PT: 16 male, 8 female; chi^2^ test: n.s.), age (median age (interquartile range) TO: 52 a (37 a, 59a); PT: 58 a (50 a, 62 a); Mann–Whitney U-test: n.s.), tinnitus parameters (TF (TO: 6 kHz (4 kHz, 8 kHz); PT: 6 kHz (3 kHz, 6 kHz); Mann–Whitney U-test: n.s.), and TL (TO: 5 dB SL (2 dB SL, 8 dB SL); PT: 4 dB SL (3 dB SL, 9 dB SL); Mann–Whitney U-test: n.s.)) were not significantly different between the TO and PT group. The same was true for the initial THQ score (Mann–Whitney U-test, *p* = 0.27), whose medians (interquartile ranges) were 28 (18.5, 35.5) for TO patients and 21.5 (11.5, 36.5) for PT patients. The maximum THQ score (range: 0 to 84) reached over all patients was 57 (67.9% of maximum), and the minimum was 4. Moreover, between the subgroups within the TO and PT patient groups, characterized by using either the internal noise generator of the HA or the optimal noise presented via an iPod, we found—with the exception of the age of both TO patient subgroups (chi^2^ test, *p* = 0.02)—no significant differences in those parameters (cf. Appendix A). Therefore, the results of the two patient groups could be well compared without splitting them into subgroups, while considering the difference in group size. To rule out the effects of a single subgroup, we added the appropriate analyses after the investigation of the complete patient cohort.

With this in mind, we first compared the audiograms of both patient groups by a two-factorial ANOVA with the factors *group* and *frequency* (Figure 2). We found a slight but significant difference in the mean HL (±95% CI) of both *groups*, with the TO patients showing a 14.6 dB ± 0.72 dB and the PT patients having a 17.9 dB ± 1.02 dB hearing deficit. A clinically relevant HL of more than 20 dB was only shown in *frequencies* at and above 3 kHz. No *interaction* of both factors was found (*p* = 0.24), indicating a general slightly worse hearing in the PT patients compared to the TO group.

Investigating the TF distributions in both patient groups (Figure 3A), we found no difference in the distributions of both patient groups (two-sample Kolmogorov–Smirnov test, D = 0.3, K = 0.67, *p* = 0.66). The TF was found to be relatively weakly linearly related to the maximum HL frequency (Figure 3B) in the TO patients (r^2^ = 0.24, *p* < 0.001), and we found only a trend for such a relationship in the PT patients (r^2^ = 0.09, *p* = 0.06). A much stronger relationship was found between TF and the LIN center frequency (Figure 3C), where both slopes became significant (*p* < 0.001 in both groups) and the deviation from the linear fit was relatively small (TO: r^2^ = 0.73, PT: r^2^ = 0.51). Such relationships were also found in the subgroups of both the TO and PT groups (*p* < 0.001 in all groups).

### 3.3. Efficacy of LINTS

For the investigation of the efficacy of the LINTS treatment, we evaluated the THQ scores in general (global tinnitus severity) and in the five sub-scores over time (cf. Methods and Table 1). In the nonparametric statistical analysis of the global tinnitus severity score change by the Friedman ANOVAs (Figure 4A, Table 1) for both patient groups, we found a significant temporal change only in the TO patient group (*p* = 0.035), while the PT patients did not show a significant development over time (*p* = 0.27). When assessing the change in the global tinnitus severity relative to the pre-treatment levels by the Wilcoxon tests (Figure 4A asterisks, Table 2), we find this temporal effect in the TO patient group already starting after 2 weeks of treatment (−3 (−7, 2); *p* = 0.01), peaking at 4 weeks of treatment (−4 (−9, 0); *p* < 0.001), but staying significantly lowered up to 4 weeks after the end of the treatment (−3 (−8, 1); *p* = 0.009). In this case, the NB- and (AM-)optimized LINTS stimulation seems to be superior to the WB noise stimulation, as the most beneficial effects at these three timepoints can be found from that patient group and the more increased global tinnitus severity value changes showed up in the latter subgroup (Figure 4B). This fact becomes evident when Mann–Whitney-U tests are performed between the values of the global tinnitus severity change of the subgroups of TO patients, where at least the tendencies and one significant value for this difference can be found: 2 weeks of treatment U = 254.5, *p* = 0.04; 4 weeks of treatment U = 240.0, *p* = 0.10; 4 weeks post treatment U = 236.5, *p* = 0.05 (cf. also Figure 5). As described above, to investigate the benefit of the LINTS treatment in detail, we analyzed the THQ sub-scores on their temporal development (Table 1) and the effects at each timepoint (Table 2). As an example, for the beneficial effect of the treatment in both patient groups (TO and PT), we show the development of the tinnitus intrusiveness in Figure 4C. Here, both patient groups show comparable temporal effects, with the intrusiveness (maximal score: 16) dropping significantly after four weeks of treatment (TO: −3 (−2, −5); PT −2 (0, −4)). Nevertheless, this change does not seem to be stable, as it vanished after 4 weeks after the treatment. In the TO patient group, we find a significant benefit or trend for such a temporal development in 3 out of the 5 sub-scores (emotional and cognitive distress, tinnitus intrusiveness, and auditory perceptual difficulties; Table 1) and a significant benefit or trend in 11 of the 15 sub-score timepoints (Table 2), with beneficial effects in 4 out of 5 sub-scores in the long-term effect measurement 4 weeks after treatment (all except auditory perceptual difficulties). In the PT group, the 2 weeks of white noise seemed to have a hindering effect on the treatment, as only the intrusiveness showed this temporal pattern and somatic complains showed a trend for this. On the single timepoint level, the beneficial effects are also small, as we find a significant decrease in the score only in 4 out of 20 timepoints. This difference in treatment efficacy can also be seen in the change in the tinnitus severity index in both groups (Figure 4D). While patients in the PT group hardly showed any group shift to a lower index number after 4 weeks of treatment (2/24, 8%), in the TO group, 11 out of 48 (23%) moved their tinnitus severity ranking down at least one index number (with 5 patients reporting no more tinnitus percept during stimulation) and 8 of them stayed there even 4 weeks after the end of treatment.

As mentioned above, we wanted to rule out that only one of the subgroups of the TO or PT patient groups showed the described effects (Figure 5). Therefore, the THQ score changes of each subgroup have been tested separately for a change relative to the pre-treatment values (Wilcoxon tests). We found all TO patient subgroups to show a significant decrease of 4 to 6 points (WB: *p* = 0.009; NB: *p* = 0.006, NB AM: *p* = 0.026), with the NB stimulation having the numerically strongest effect. No PT subgroup reached significance in these tests. Neither the TO nor PT subgroups differed significantly from each other (Kruskal–Wallis ANOVA, *p* > 0.05 in both cases). In other words, the subgroups showed comparable effects and were rightfully taken together as single patient groups.

As the distribution of the tinnitus severity index in Figure 4D showed, the efficacy of the treatment might be obscured by the relatively large number of patients with a tinnitus severity index of I. Therefore, we calculated the percentage change in the general tinnitus severity score (Figure 6) and performed two repeated-measurement ANOVAs over the *timepoints* for both groups (due to the different number of measurements, a combination of both groups was not possible). We found a comparable effect in the TO patient group with a significant temporal effect (*p* = 0.04) and a significant decrease (*t*-test, *p* < 0.001) in the global tinnitus severity by −17.1% ± 6.9% after 4 weeks of LINTS treatment that stayed decreased at −11.4% ± 7.6% 4 weeks after the treatment. For the PT patients, we also found a temporal effect (*p* = 0.04). Nevertheless, we could not identify a significant timepoint but only found a beneficial trend at 4 weeks of treatment (−8.5% ± 12.0%, *p* = 0.09). In other words, this significant temporal effect indicates that the high number of low-tinnitus-severity-index patients is indeed decreasing the nonparametric analysis power at least in PT patients. This is because small numerical improvements in less severely affected patients result in a small absolute change in numbers, but a relatively large percentage change. Nevertheless, the overall effect had a tendency to be stronger without the preceding white noise placebo treatment, with the mean change (±standard deviation) during and after treatment in TO patients at −10.9% ± 25.8% and in PT patients at −4.7% ± 30.8% (*t*-test, *p* = 0.08). The peak of the benefit was found at the end of the treatment period in both patient groups with TO at −17.1% ± 22.6% and PT at −8.6% ± 29.5%, these values also showed a trend to be different (*t*-test, *p* = 0.07).

### 3.4. Exemplary Subjective Patient Reports

As an exemplary patient for a successful LINTS treatment, we want to present a 52-year-old male tinnitus patient with a mean HL of 36.5 dB and 25.3 dB in the left and right ear, respectively. He perceived a TF of 6 kHz and a TL of 5 dB SL in the left ear and reported hearing it for 4.4 years with an unknown origin. He started with a tinnitus severity index of I (mildly affected, THQ score 17). He was provided with an HA with the internal noise generator turned on and placed in the TO group. He reported a complete silencing of tinnitus during the wearing of the HA. But, especially during the treatment period, he was more aware of his tinnitus when not wearing it and reporting even some problems in going to sleep, as he was more attentive to the percept during the quiet night. His THQ score dropped only by 11.8% after 2 weeks and 17.6% after 4 weeks of treatment. Nevertheless, after the end of the active treatment period, he reported, after 4 weeks, a THQ score that was 47.1% lower (THQ score 8) than at the beginning. Especially, the emotional and cognitive distress and the tinnitus intrusiveness scores were diminished by half. Even though he never switched the tinnitus severity index, when asked in the end, he reported that he was relieved of being bothered by any tinnitus percept.

Another example for a successful LINTS treatment of a patient affected by more severe tinnitus is the case of a 42-year-old male patient with a mean HL of 16.5 dB and 8.0 dB in the left and right ear, respectively. He perceived a TF of 4 kHz in both ears at 7 and 2 dB SL, respectively, and reported hearing it since roughly 6 years ago after a stressful episode in his life. He started with a tinnitus severity index of III (severe, THQ score 50). He was provided with an HA without the internal noise generator—i.e., the optimal noise was streamed via an iPod—and placed in the TO group. He reported a significant improvement in his tinnitus loudness and also wore the HAs during the night, as he was then able to fall asleep without the tinnitus bothering him. His THQ score dropped by 25% after 2 weeks and 40% (I, mildly affected, THQ score 30) after 4 weeks of treatment. After 4 weeks without the treatment, his score was only 12% below the starting level. He asked to receive the stimulation files to play it via his own in-ear devices. After receiving it, he reported back after a few days that he can sleep again and is now able to cope with his tinnitus.

## 4. Discussion

In this Phase II clinical study, we wanted to further investigate the efficacy of LINTS in suppressing phantom perception in patients with subjective tinnitus. We were able to show that LINTS can significantly reduce the THQ score, with certain sub-scores, such as intrusiveness, benefiting more than others (cf. Figure 4). At first glance, the average reduction in the THQ score of 4 after four weeks of treatment does not appear particularly strong, but one has to keep in mind that the average median THQ score of our patients was 26.5 and, therefore, the average relative improvement was 15%. Nevertheless, the classic 12-points improvement for “clinical relevance”, i.e., the 15% of the full score, was not reached in most patients (cf. Figure 4B). Certain sub-scores which are particularly decisive for the subjectively perceived psychological strain, like intrusiveness, fortunately showed the strongest effects (cf. Table 1 and Table 2) on the group level. Additionally, we observed strong inter-individual differences in the effect LINTS had on tinnitus patients: while some patients did not seem to profit from the treatment, LINTS was able to subjectively suppress the tinnitus percept in others completely (cf. subjective patient reports). Furthermore, the THQ is more suited for long-term effects [31], even though we explicitly instructed the patients to report the effects during the last two or four weeks, respectively. Therefore, the reported effects could be an underestimation, obscured by the long-term condition the patients suffer from.

An important observation is the reduced effect of LINTS found in the PT patients. The “wrong” stimulation in the first two weeks with low-intensity white noise—although barely perceivable in normal sound environments—did not show the classic “placebo effect” [38], which leads to improvements solely due to the intention of treatment, but showed no change at all in the THQ scores. This was also the main reason for dropouts in this group. The following “correct” stimulation in the PT group showed a strong delay in its effect compared to the TO group, which seemed to start only after the end of the four-week treatment period. Therefore, it is worth discussing if this was a true placebo approach, as the WN obviously had a negative effect on our LINTS approach. This is in line with the reports on the maladaptive effects of WN in tinnitus patients [39], even though the intensity of the used stimulations differ significantly.

In addition to the relatively small number of patients, another drawback of this study is the usage of two different noise-generating approaches, which might add to the inter-individual differences found, even though we do not see this on the group level (cf. Figure 5). Strong inter-individual differences in the effectiveness of tinnitus treatment have also been reported for other approaches that are based on acoustic stimulation (e.g., tailor-made notched music therapy [13] or coordinated reset [14], which, on the other hand, is even recommended to be avoided in tinnitus patients [9]). To our knowledge, LINTS is currently the only acoustic-stimulation-based therapeutic approach for treating tinnitus whose effectiveness has not yet been disproved (like, e.g., [14,15,16,40]). Actually, the general use of tones, auditory scenes, and broadband or narrowband noise in the tinnitus frequency range has been tested in many approaches and forms of application for tinnitus treatment; none of the procedures could be proven effective, or studies on this topic were not initiated at all [13,14,41,42,43,44,45,46,47,48,49,50,51,52,53,54,55,56,57]. Moreover, treatment with maskers were not shown to be effective, except that one masks one sound with another [58]. We would, therefore, like to encourage colleagues to take over our approach and try to independently falsify or replicate our results! Furthermore, approaches using EEG, MEG, or other objective measurements of brain states to search for an objective correlate of a therapeutic effect induced by LINTS are highly encouraged. Please note that LINTS differs fundamentally from the aforementioned classic tinnitus sound therapies, as the amplitude of the sound used is significantly below the masking threshold (cf. [59]) and the frequency spectrum is individually adjusted to the tinnitus pitch perceived by the patient. Interestingly, several patients report in personal communication that they use LINTS already in some form of self-treatment. They play very soft sounds from their mobile devices to lower their tinnitus intensity without masking it. In that sense, LINTS is already widely used, but not in a controlled manner.

We believe that the main reason for the large inter-individual differences in the effectiveness of LINTS lies in the difficulty of optimally adjusting the noise to the patient’s individual tinnitus perception and hearing threshold. On the one hand, there is the difficulty of precisely determining the pitch of tinnitus: since this cannot be measured objectively but only approximated subjectively, and octave errors can also occur, this is the greatest source of error in the adjustment of LINTS. On the other hand, an individual’s tinnitus perception can also change over time, meaning that regular adjustments to the stimulation may be necessary. We are, therefore, convinced that, with the optimal adjustment of the stimulation to the tinnitus perception, it should be possible to completely suppress tinnitus subjectively in many patients, at least in those who have a tonal percept and are able to hear at least to a certain degree. Even if the percept can “only” be tuned down to a more bearable level, this regaining of control would be most valuable for individual patients. The fact that this adjustment involves a number of parameters, such as the center frequency of the bandpass noise, the spectral width of the bandpass, and the absolute volume, makes the process potentially time-consuming and, therefore, sometimes only acceptable to patients to a limited extent. At least, the additional amplitude modulation did not appear to improve the therapeutic outcome (cf. Figure 5), so it does not seem necessary to pursue this approach further in the future. Nevertheless, optimizing this process and making it as time-efficient as possible will be one of the key tasks in the future development of this therapeutic approach.

Another potential problem for the success of the therapy—as with any therapy—is the issue of patient compliance: it stands to reason that the success of the therapy also depends on the duration of stimulation per day. The minimum wearing time of 4 h per day for hearing aids is certainly not optimal in this case, which contributed to the inter-individual differences. Nevertheless, most patients wore the HA significantly longer than the minimum time and, in some cases, even during sleep to prevent a rebound of the tinnitus percept after removing the HAs and lying in a quiet environment.

Finally, the device used for stimulation will have an impact on the success of the therapy, as technical limitations can prevent the parameters for LINTS from being implemented, even if they have been optimally determined. This was also the case in the present study at the beginning, when we only worked with the degrees of freedom of the hearing aid itself. After switching to stimulation with iPods, this problem was solved, which then also improved the success of the therapy (cg. Figure 5, WB vs. NB).

## 5. Conclusions

LINTS has been shown to be a promising therapeutic approach for the partial or subjectively complete suppression of tinnitus in our clinic. Nevertheless, it can only be seen as a preliminary report, needing further research for optimization and validation. The quality of the stimulation adjustment to the individual hearing loss and the patient’s perception of tinnitus appears to be particularly critical for the effectiveness of the therapy. In order for LINTS to become established in everyday clinical practice, the independent validation of the procedure and the optimization of the fitting process are necessary.

## Figures and Tables

**Figure 1 brainsci-15-01222-f001:**
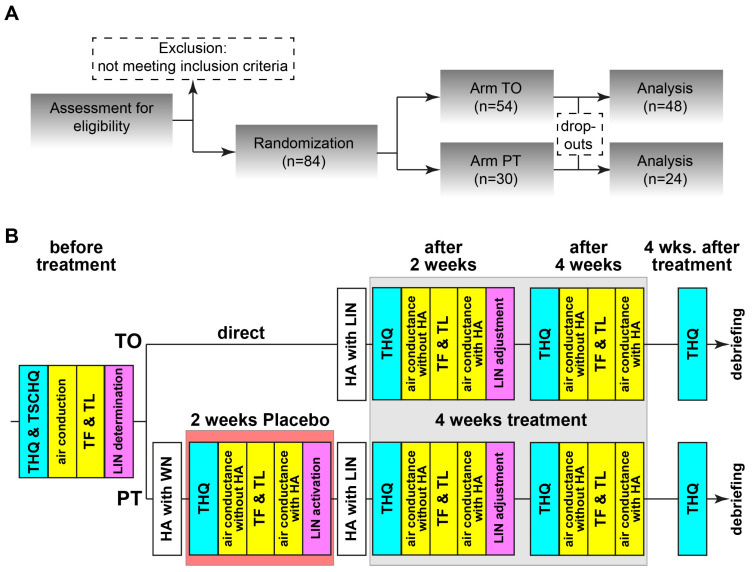
Flow diagram and timeline of the study. (**A**) All patients were investigated according to this flow chart. TO: noise-treatment-only, PT: placebo-plus-noise-treatment. (**B**) Details of the two therapeutic arms. THQ: tinnitus health questionnaire, TSCHQ: Tinnitus Sample Case History Questionnaire, TF: tinnitus frequency, TL: tinnitus loudness, LIN: low-intensity noise, HA: hearing aid.

**Figure 2 brainsci-15-01222-f002:**
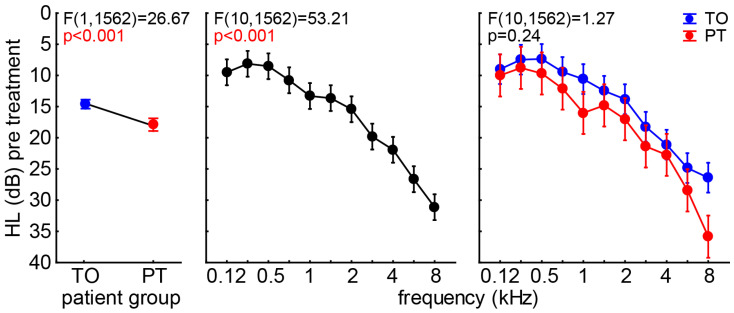
Two-factorial ANOVA results of patients’ audiograms. From left to right, the three results of the analysis are given for factor *group*, *frequency*, and the *interaction* of both factors. Blue symbols represent the means ± 95% confidence interval (CI) of the TO patients, and red the ones for the PT patient group.

**Figure 3 brainsci-15-01222-f003:**
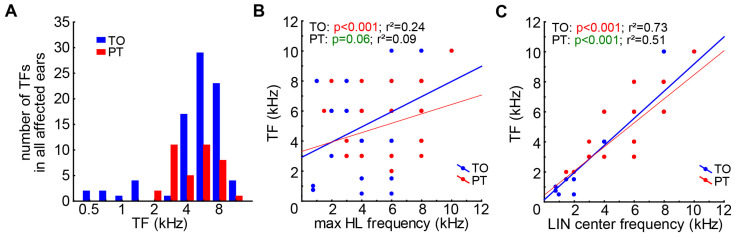
TF distributions and correlations. (**A**) Distributions of TFs of both patient groups. (**B**) Linear correlation and regression coefficient of frequency with strongest HL and TF of TO patients, with a trend for such a linear correlation for PT patients. (**C**) Significant correlations and regression coefficients of the LIN center frequency and the TF in both patient groups.

**Figure 4 brainsci-15-01222-f004:**
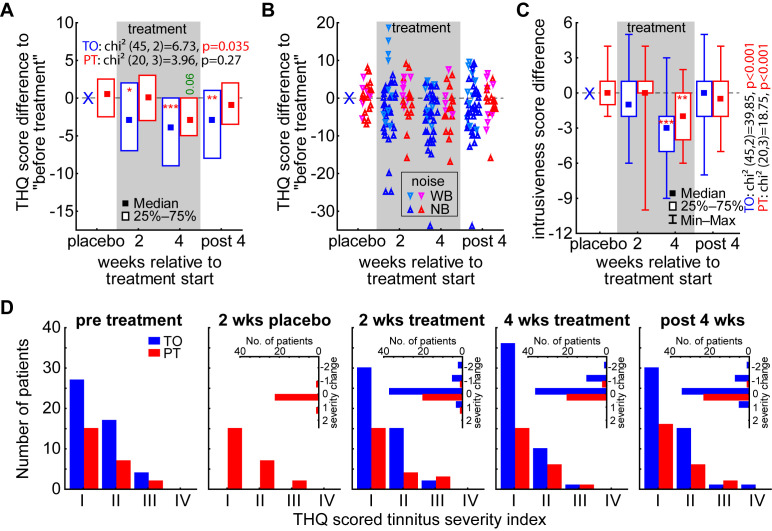
Nonparametric analyses of THQ scores and tinnitus severity index. (**A**) Results of the two Friedman ANOVAs of the THQ score difference over time in both patient groups. The median and the interquartile range are given as boxes. Asterisks indicate significant differences to pre-treatment scores (Wilcoxon tests). Green value: trend, * *p* < 0.05, ** *p* < 0.01, *** *p* < 0.001. (**B**) Violin plots of the patients’ score differences with separation of the TO (bluish colors) and PT patients (reddish colors) stimulated with WB noise and NB noise. (**C**) Results of the two Friedman ANOVAs of the intrusion sub-score. Median, interquartile range, and full distribution (whiskers) are given. Asterisks as above. (**D**) Distributions of tinnitus severity indices obtained from the THQ scoring for all timepoints. Insets show the change in index to lower (negative numbers) or higher indices for the patients separated by group.

**Figure 5 brainsci-15-01222-f005:**
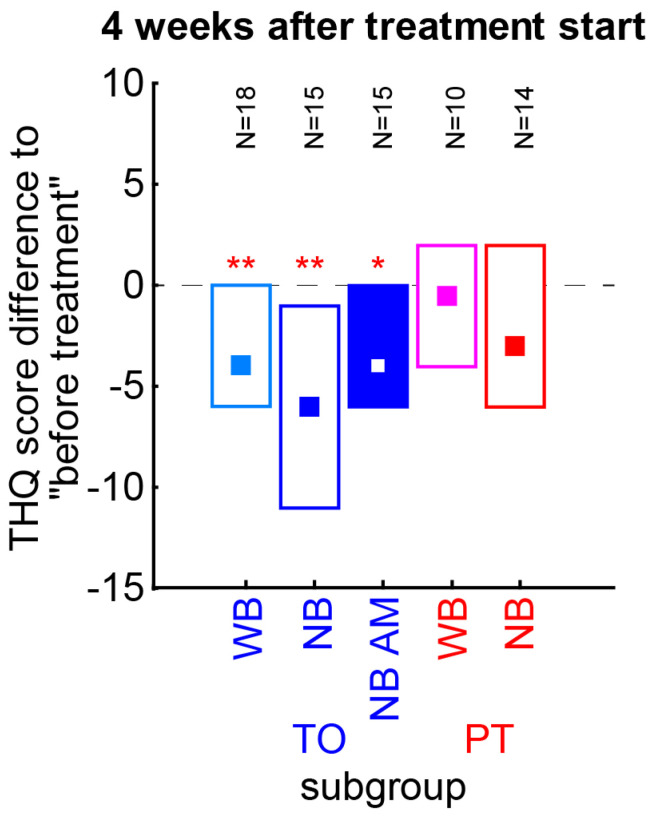
Nonparametric analyses of THQ scores of TO and PT patient subgroups four weeks after treatment start. TO patients can be divided into WB, and NB without or with AM-modulated noise, and PT patients into WB and NB noise subgroups. The number of patients in each group is given above the median values with interquartile ranges (boxes). Colors as in Figure 4B. Asterisks indicate a significant improvement relative to pre-treatment values (Wilcoxon test). Since the scores do not differ significantly between the NO subgroups or between the PN subgroups, the summary of the data as presented in the above analyses is justified. * *p* < 0.05, ** *p* < 0.01.

**Figure 6 brainsci-15-01222-f006:**
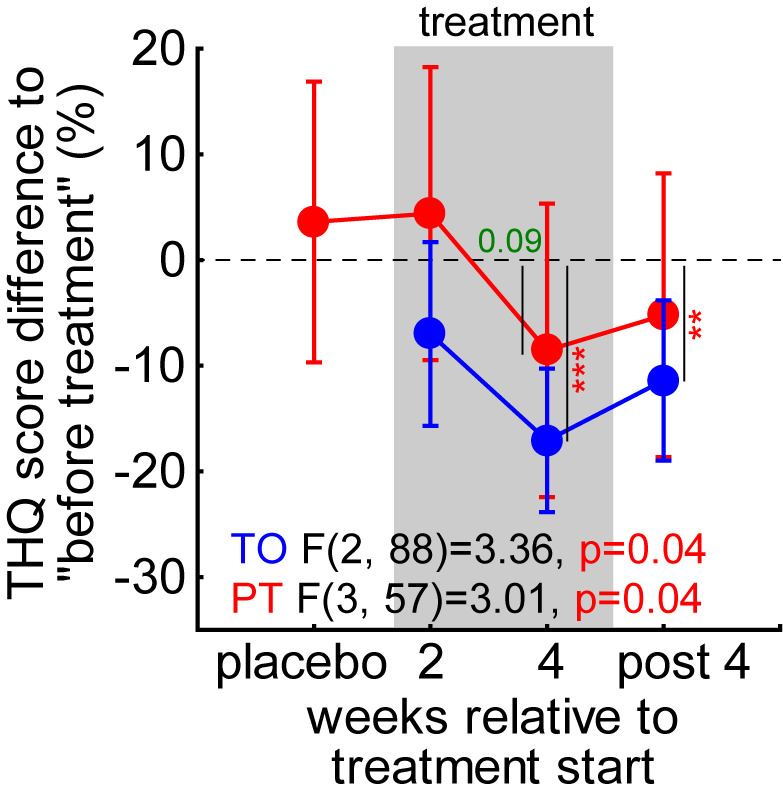
One-factorial ANOVAs of percentual change in THQ score. Two independent significant ANOVAs were calculated for both patient groups. Vertical lines with numbers/asterisks give the timepoints of significant decreases relative to pre-treatment (t-tests). Green value: trend, ** *p* < 0.01, *** *p* < 0.001.

**Table 1 brainsci-15-01222-t001:** Friedman ANOVA results for THQ sub-scores and global tinnitus severity changes of both patient groups over time.

Group	(Sub-)Score	Timepoint	chi^2^ Test; Effect Size	Score Means	Standard Deviation
treatment-only patient group (TO)	emotional and cognitive distress	2 weeks of treatment	(N = 45. df = 2) = 10.47 *p* = 0.005 W = 0.04; *weak*	−1.33	4.82
		4 weeks of treatment		−2.87	4.21
		4 weeks post treatment		−1.8	4.63
	intrusiveness	2 weeks of treatment	(N = 45. df = 2) = 39.85 *p* < 0.001 W = 0.37; *strong*	−0.62	2.57
		4 weeks of treatment		−3.36	2.45
		4 weeks post treatment		−0.60	2.21
	auditory perceptual difficulties	2 weeks of treatment	(N = 45. df = 2) = 5.15 *p* = 0.076 W = 0.02; *weak*	−0.07	1.74
		4 weeks of treatment		−0.36	1.88
		4 weeks post treatment		−0.18	2.24
	sleep disturbances	2 weeks of treatment	(N = 45. df = 2) = 0.57 *p* = 0.75 W = 0.01	−0.42	1.14
		4 weeks of treatment		−0.40	1.32
		4 weeks post treatment		−0.49	0.97
	somatic complains	2 weeks of treatment	(N = 45. df = 2) = 0.64 *p* = 0.73 W = 0.06	−0.29	1.12
		4 weeks of treatment		−0.36	1.13
		4 weeks post treatment		−0.47	1.12
	global tinnitus severity	2 weeks of treatment	(N = 45. df = 2) = 6.73 *p* = 0.035 W = 0.02; *weak*	−2.96	8.21
		4 weeks of treatment		−4.96	6.65
		4 weeks post treatment		−3.2	7.59
placebo-plus-treatment patient group (PT)	emotional and cognitive distress	2 weeks of placebo	(N = 20. df = 3) = 2.16 *p* = 0.54 W = 0.10	−0.40	3.86
		2 weeks of treatment		−1.20	2.44
		4 weeks of treatment		−1.65	2.35
		4 weeks post treatment		−1.35	1.90
	intrusiveness	2 weeks of placebo	(N = 20. df = 3) = 18.75 *p* < 0.001 W = 0.68; *strong*	−0.50	2.09
		2 weeks of treatment		−2.05	2.39
		4 weeks of treatment		0.05	2.94
		4 weeks post treatment		0.55	1.70
	auditory perceptual difficulties	2 weeks of placebo	(N = 20. df = 3) = 1.00 *p* = 0.80 W = 0.13	0.35	1.18
		2 weeks of treatment		0.10	1.71
		4 weeks of treatment		−0.25	1.71
		4 weeks post treatment		−0.05	1.82
	sleep disturbances	2 weeks of placebo	(N = 20. Df = 3) = 2.58 *p* = 0.46 W = 0.03	0.10	1.21
		2 weeks of treatment		0.35	1.60
		4 weeks of treatment		0.35	1.27
		4 weeks post treatment		0.70	2.27
	somatic complains	2 weeks of placebo	(N = 20. df = 3) = 7.68 *p* = 0.053 W = 0.22; *moderate*	0.65	1.84
		2 weeks of treatment		0.20	0.83
		4 weeks of treatment		−0.15	0.67
		4 weeks post treatment		0.10	0.79
	global tinnitus severity	2 weeks of placebo	(N = 20. df = 3) = 3.96 *p* = 0.27 W = 0.09	0.55	4.39
		2 weeks of treatment		−0.55	5.87
		4 weeks of treatment		−2.35	5.84
		4 weeks post treatment		−1.75	5.00

Note: Red-colored *p*-values indicate a significant effect of the treatment over time; green-colored *p*-values indicate a trend. Effect size and (if *p* < 0.1), its interpretation, are given as Kendall coefficient of concordance (Kendall’s W).

**Table 2 brainsci-15-01222-t002:** Wilcoxon matched-pair tests of THQ sub-scores and global tinnitus severity changes of both patient groups relative to pre-treatment level.

Group	(Sub-)Score	Timepoint	Wilcoxon Test; Effect Size	Median (25%, 75%)
treatment-only patient group (TO)	emotional and cognitive distress	2 weeks of treatment	T = 252.5; Z = 2.12 *p* = 0.03; η^2^ = 0.11; *weak*	−1 (−3, 1)
		4 weeks of treatment	T = 136.5; Z = 3.68 *p* < 0.001; η^2^ = 0.34; *fair*	−2 (−4, 0)
		4 weeks post treatment	T = 250; Z = 2.34 *p* = 0.02; η^2^ = 0.13; *weak*	−1 (−4, 1)
	intrusiveness	2 weeks of treatment	T = 331; Z = 1.29 *p* = 0.20; η^2^ = 0.04	−1 (−2, 1)
		4 weeks of treatment	T = 29; Z = 5.51 *p* < 0.001; η^2^ = 0.68; *moderate*	−3 (−5, −2)
		4 weeks post treatment	T = 214.5; Z = 1.86 *p* = 0.06; η^2^ = 0.10; *weak*	0 (−2, 1)
	auditory perceptual difficulties	2 weeks of treatment	T = 250.5; Z = 0.25 *p* = 0.80; η^2^ = 0.002	0 (−1, 1)
		4 weeks of treatment	T = 245; Z = 1.38 *p* = 0.17; η^2^ = 0.05	0 (−1, 1)
		4 weeks post treatment	T = 216; Z = 0.34 *p* = 0.73; η^2^ = 0.004	0 (−1, 1)
	sleep disturbances	2 weeks of treatment	T = 138; Z = 1.94 *p* = 0.05; η^2^ = 0.13; *weak*	0 (−1, 0)
		4 weeks of treatment	T = 131; Z = 1.87 *p* = 0.06; η^2^ = 0.12; *weak*	0 (−1, 0)
		4 weeks post treatment	T = 47.5; Z = 2.56 *p* = 0.01; η^2^ = 0.30; fair	0 (−1, 0)
	somatic complains	2 weeks of treatment	T = 61; Z = 2.34 *p* = 0.02; η^2^ = 0.24; fair	0 (−1, 0)
		4 weeks of treatment	T = 56; Z = 2.07 *p* = 0.04; η^2^ = 0.20; fair	0 (−1, 0)
		4 weeks post treatment	T = 44; Z = 1.81 *p* = 0.07; η^2^ = 0.18; *weak*	0 (−1, 0)
	global tinnitus severity	2 weeks of treatment	T = 267; Z = 2.49 *p* = 0.01; η^2^ = 0.14; *weak*	−3 (−7, 2)
		4 weeks of treatment	T = 120.5; Z = 4.37 *p* < 0.001; η^2^ = 0.43; *moderate*	−4 (−9, 0)
		4 weeks post treatment	T = 285.5; Z = 2.62 *p* = 0.009; η^2^ = 0.15; *weak*	−3 (−8, 1)
placebo-plus-treatment patient group (PT)	emotional and cognitive distress	2 weeks of placebo	T = 64; Z = 1.25 *p* = 0.21; η^2^ = 0.08	−0.5 (−3, 1.5)
		2 weeks of treatment	T = 28; Z = 2.30 *p* = 0.02; η^2^ = 0.31; *fair*	−1 (−3, 0)
		4 weeks of treatment	T = 31.5; Z = 2.56 *p* = 0.01; η^2^ = 0.35; *fair*	−2 (−2, −1)
		4 weeks post treatment	T = 16.5; Z = 2.66 *p* = 0.008; η^2^ = 0.44; *moderate*	−0.5 (−3, 0)
	intrusiveness	2 weeks of placebo	T = 65; Z = 0.89 *p* = 0.37; η^2^ = 0.04	0 (−1, 1)
		2 weeks of treatment	T = 45; Z = 0.85 *p* = 0.39; η^2^ = 0.05	0 (0,1)
		4 weeks of treatment	T = 13; Z = 3.16 *p* = 0.002; η^2^ = 0.55; *moderate*	−2 (−4, 0)
		4 weeks post treatment	T = 85; Z = 0.75 *p* = 0.46; η^2^ = 0.06	−0.5 (−2, 1)
	auditory perceptual difficulties	2 weeks of placebo	T = 65; Z = 0.54 *p* = 0.59; η^2^ = 0.02	0 (−1, 1)
		2 weeks of treatment	T = 63.5; Z = 0.23 *p* = 0.82; η^2^ = 0.003	0 (−1, 1)
		4 weeks of treatment	T = 49.5; Z = 0.96 *p* = 0.34; η^2^ = 0.06	0 (−2, 1)
		4 weeks post treatment	T = 69; Z = 0.36 *p* = 0.72; η^2^ = 0.008	0 (−1, 1)
	sleep disturbances	2 weeks of placebo	T = 27.5; Z = 1.26 *p* = 0.21; η^2^ = 0.12	0 (0, 1)
		2 weeks of treatment	T = 20; Z = 1.16 *p* = 0.25; η^2^ = 0.12	0 (0, 1)
		4 weeks of treatment	T = 33; Z = 0.47 *p* = 0.64; η^2^ = 0.02	0 (−1, 0)
		4 weeks post treatment	T = 29; Z = 0.36 *p* = 0.72; η^2^ = 0.01	0 (−1, 0)
	somatic complains	2 weeks of placebo	T = 12; Z = 1.58 *p* = 0.11; η^2^ = 0.25	0 (0, 1)
		2 weeks of treatment	T = 10.5; Z = 1.42 *p* = 0.16; η^2^ = 0.22	0 (0, 1)
		4 weeks of treatment	T = 6; Z = 0.94 *p* = 0.35; η^2^ = 0.15	0 (0, 0)
		4 weeks post treatment	T = 8; Z = 1.01 *p* = 0.31; η^2^ = 0.15	0 (0, 0)
	global tinnitus severity	2 weeks of placebo	T = 106.5; Z = 0.31 *p* = 0.75; η^2^ = 0.005	0.5 (−2.5, 2.5)
		2 weeks of treatment	T = 83; Z = 0.11 *p* = 0.91; η^2^ = 0.0007	0 (−3, 3)
		4 weeks of treatment	T = 55; Z = 1.87 *p* = 0.06; η^2^ = 0.17; *weak*	−3 (−5, 0)
		4 weeks post treatment	T = 88; Z = 1.25 *p* = 0.21; η^2^ = 0.07	−1 (−3.5, 2)

Note: Red-colored *p*-values indicate a significant difference in the median value relative to pre-treatment; green-colored *p*-values indicate a trend. Effect size and (if *p* < 0.1), its interpretation, are given as η^2^ values.

## Data Availability

The raw data supporting the conclusions of this article will be made available by the authors, without undue reservation.

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
