# Peer review of "Phase II Clinical Study on Low-Intensity-Noise Tinnitus Suppression (LINTS) for Tinnitus Treatment"

_brainsci, 2025, doi:10.3390/brainsci15111222_

Round 1

Reviewer 1 Report

Comments and Suggestions for Authors

This paper investigates the use of individualized non-masking low intensity noise (LIN) on tinnitus. Data from two groups are presented, one where participants received treatment only over four weeks and another where they received placebo sound for two weeks, followed by the treatment for four weeks. In the treatment only group, a reduction in the THQ score after two and four weeks was observed. In the placebo and treatment group, less effectiveness was observed.

One question is around the design of the study. The rationale behind having a treatment only group and a placebo plus treatment group is unclear. Wouldn’t a single placebo plus treatment group, where the order of placebo or treatment was counterbalanced, be more effective in answering the research question?

There is not much discussion on why the authors think the placebo plus treatment group didn’t see much effect.

Under Statistical Evaluation, what do the authors mean by ‘detail analyses’ (line 187)? Are they referring to pairwise comparisons?

Also, why were non-parametric statistics used only? There is no mention of testing assumptions but with the sample sizes included, it seems unlikely that all tests needed to be non-parametric. Also, in the Results section there is mention of ANOVAs which doesn’t agree with the Methods.

Line 197: there is mention of excluding patients with highest severity. How many patients were excluded and why was it decided to exclude them?

Line 216: what do the authors mean by self-developed computer program?

There is repetition of Methods under Results which is unnecessary (in particular Study Design under Results is mostly Methods).

Line 441: the authors claim that their approach is the only acoustic stimulation-based approach not disproved. Do they have evidence that all other methods (not just coordinated reset) have been shown to not be effective? If so, they need to provide references.

Line 454: the authors suggest that with optimal adjustments tinnitus can be completely suppressed. That is a big claim and the results presented do not provide enough evidence to back this claim.

Author Response

One question is around the design of the study. The rationale behind having a treatment only group and a placebo plus treatment group is unclear. Wouldn’t a single placebo plus treatment group, where the order of placebo or treatment was counterbalanced, be more effective in answering the research question?

The reason for this approach was that we wanted to investigate the possible long-term effects after the end of LINTS therapy without any interference. Therefore, we refrained from a “balanced” approach with the TO group receiving a WN stimulation after the end of the treatment. Furthermore, we wanted to control for “any” stimulation effect with the WN stimulation in the first two weeks in the PT group. We stated this in L328ff

There is not much discussion on why the authors think the placebo plus treatment group didn’t see much effect.

This is now added to the Discussion section L725ff

Under Statistical Evaluation, what do the authors mean by ‘detail analyses’ (line 187)? Are they referring to pairwise comparisons?

L407: It is now clearly stated, that pairwise test have been used

Also, why were non-parametric statistics used only? There is no mention of testing assumptions but with the sample sizes included, it seems unlikely that all tests needed to be non-parametric. Also, in the Results section there is mention of ANOVAs which doesn’t agree with the Methods.

We added the information to the statistics section L414ff

Line 197: there is mention of excluding patients with highest severity. How many patients were excluded and why was it decided to exclude them?

L430: We clarified this misunderstanding. We did not include patients with a severity higher than 3.

Line 216: what do the authors mean by self-developed computer program?

L369: We clarified, that it is a in-house developed Python program.

There is repetition of Methods under Results which is unnecessary (in particular Study Design under Results is mostly Methods).

We removed the repeating section from the results.

Line 441: the authors claim that their approach is the only acoustic stimulation-based approach not disproved. Do they have evidence that all other methods (not just coordinated reset) have been shown to not be effective? If so, they need to provide references.

L714ff: We added more details to this part of the Discussion

Line 454: the authors suggest that with optimal adjustments tinnitus can be completely suppressed. That is a big claim and the results presented do not provide enough evidence to back this claim.

We rephrased our interpretation and stated, that it may reduce the percept in many patients and may even remove it completely. L782ff

Reviewer 2 Report

Comments and Suggestions for Authors

The authors present a double-blind, placebo-controlled Phase II clinical trial evaluating the efficacy of Low Intensity Noise Tinnitus Suppression (LINTS) in 72 adult tinnitus patients. The treatment involves individualized, non-masking low-intensity noise (LIN) delivered via hearing aids. Patients were assigned to either a treatment-only (TO) group or a placebo-plus-treatment (PT) group. Outcomes were measured using the Tinnitus Health Questionnaire (THQ) and its subscales.

Here are the major concerns:

The study is built on the Erlangen model of tinnitus, which posits that tinnitus arises from stochastic resonance (SR) in the dorsal cochlear nucleus (DCN). While intriguing, this model remains largely speculative and lacks robust empirical validation. The authors extrapolate from this model to propose LINTS, yet no direct neurophysiological evidence is presented to confirm that externally applied low-intensity noise modulates SR in the DCN or suppresses tinnitus via this mechanism. The claim that LINTS can “completely suppress tinnitus” in some patients is scientifically irresponsible without objective verification (e.g., neuroimaging, electrophysiology).

The study includes 72 patients (TO: 48, PT: 24), but no power analysis is provided to justify whether this sample size is sufficient to detect clinically meaningful effects. The lack of power analysis undermines the reliability of the statistical conclusions, especially given the modest effect sizes and high interindividual variability.

Blinding was compromised after the third measurement, as investigators needed to know group allocation to schedule further sessions. This introduces potential observer bias, especially in subjective outcome measures like THQ.

The PT group received white noise instead of LIN for the first two weeks. However, white noise is not a true inert placebo—it may have auditory effects. This could confound the interpretation of treatment efficacy and reduce internal validity.

The authors use a mix of parametric (ANOVA, t-tests) and nonparametric (Wilcoxon, Friedman ANOVA) tests without clearly justifying the choice or checking assumptions (e.g., normality, homogeneity). This raises concerns about statistical rigor and the appropriateness of the analyses.

Multiple Wilcoxon tests are performed across timepoints and subscales, but no correction (e.g., Bonferroni, Holm) is applied. This inflates the risk of Type I errors, making some reported significant findings potentially spurious.

The study reports p-values but does not report effect sizes (e.g., Cohen’s d, η²), which are essential for interpreting clinical relevance. Without effect sizes, it is difficult to assess whether statistically significant changes are clinically meaningful.

Subgroup comparisons (e.g., WB vs. NB vs. AM noise) are conducted with small sample sizes and without pre-registration. These analyses are exploratory and should be interpreted cautiously. The authors overstate their implications.

The THQ is used as the primary outcome, but its sensitivity to short-term changes is questionable. The authors acknowledge this but still rely heavily on it. This may underestimate or misrepresent treatment effects, especially for short-term outcomes.

Variables such as tinnitus duration, age, and baseline severity are mentioned but not adequately controlled in the analysis. These could confound the results and obscure true treatment effects.

The authors claim that LINTS can “completely suppress tinnitus” in some cases and suggest it could work for all patients with tonal tinnitus. These claims are overstated, especially given the modest average improvements and lack of long-term follow-up.

Device Limitations: The use of hearing aids with suboptimal noise generators (WB vs. NB) introduces variability in treatment fidelity.

Compliance Monitoring: While wearing time is logged, its impact on outcomes is not statistically modeled.

Missing Data Handling: Dropouts are described, but no imputation or sensitivity analysis is conducted.

While the study presents a novel and potentially promising approach to tinnitus treatment, its methodological and statistical limitations significantly weaken the strength of its conclusions. The findings should be considered preliminary, and further research with more rigorous design and analysis is essential before LINTS can be recommended as a clinical intervention.

Author Response

The study is built on the Erlangen model of tinnitus, which posits that tinnitus arises from stochastic resonance (SR) in the dorsal cochlear nucleus (DCN). While intriguing, this model remains largely speculative and lacks robust empirical validation. The authors extrapolate from this model to propose LINTS, yet no direct neurophysiological evidence is presented to confirm that externally applied low-intensity noise modulates SR in the DCN or suppresses tinnitus via this mechanism. The claim that LINTS can “completely suppress tinnitus” in some patients is scientifically irresponsible without objective verification (e.g., neuroimaging, electrophysiology).

We agree that an objective measurement is needed to proof the concept of LINTS. We added this to the Discussion, L764

The study includes 72 patients (TO: 48, PT: 24), but no power analysis is provided to justify whether this sample size is sufficient to detect clinically meaningful effects. The lack of power analysis undermines the reliability of the statistical conclusions, especially given the modest effect sizes and high interindividual variability.

We added the power analysis results, L144ff

Blinding was compromised after the third measurement, as investigators needed to know group allocation to schedule further sessions. This introduces potential observer bias, especially in subjective outcome measures like THQ.

We agree, that the last two weeks of treatment in the PT group was not blinded anymore. This is already stated in the manuscript. Nevertheless, all patients filled out the THQs before this knowledge was given to the investigator. And the main effects are found in the TO group, where no knowledge was present at all. The THQs were evaluated without any changes to the scoring. Therefore, our main result is without any observer bias.

The PT group received white noise instead of LIN for the first two weeks. However, white noise is not a true inert placebo—it may have auditory effects. This could confound the interpretation of treatment efficacy and reduce internal validity.

You are absolutely right. The Initial assumption, that low intensity WN does not have negative effects on the patients but act as a kind of reference stimulation (L309ff) is false. We discussed this now in detail L690ff.

The authors use a mix of parametric (ANOVA, t-tests) and nonparametric (Wilcoxon, Friedman ANOVA) tests without clearly justifying the choice or checking assumptions (e.g., normality, homogeneity). This raises concerns about statistical rigor and the appropriateness of the analyses.

We added this information to the Methods section. The HL was assessed parametrical. All other discrete values were assessed non-parametrical. L396ff

Multiple Wilcoxon tests are performed across timepoints and subscales, but no correction (e.g., Bonferroni, Holm) is applied. This inflates the risk of Type I errors, making some reported significant findings potentially spurious.

We forgot to mention this. It is now stated in L399.

The study reports p-values but does not report effect sizes (e.g., Cohen’s d, η²), which are essential for interpreting clinical relevance. Without effect sizes, it is difficult to assess whether statistically significant changes are clinically meaningful.

We added the Effect size to our results. L400ff, Table 1 and 2.

Subgroup comparisons (e.g., WB vs. NB vs. AM noise) are conducted with small sample sizes and without pre-registration. These analyses are exploratory and should be interpreted cautiously. The authors overstate their implications.

We agree, that the sub-groups are small. We never wanted to overstate these results. We would like to keep them as is.

The THQ is used as the primary outcome, but its sensitivity to short-term changes is questionable. The authors acknowledge this but still rely heavily on it. This may underestimate or misrepresent treatment effects, especially for short-term outcomes.

We added this to the Discussion. L688f

Variables such as tinnitus duration, age, and baseline severity are mentioned but not adequately controlled in the analysis. These could confound the results and obscure true treatment effects.

We investigated possible effects of tinnitus duration, age and severity on the outcome, but did not see any effects, therefore we di not include this in the manuscript, as other reviewers already complained about the number of results.

The authors claim that LINTS can “completely suppress tinnitus” in some cases and suggest it could work for all patients with tonal tinnitus. These claims are overstated, especially given the modest average improvements and lack of long-term follow-up.

We reduced this claim and rephrased appropriately.

Device Limitations: The use of hearing aids with suboptimal noise generators (WB vs. NB) introduces variability in treatment fidelity.

This is a confound that is again stated in the Discussion, L701ff

Compliance Monitoring: While wearing time is logged, its impact on outcomes is not statistically modeled.

As the difference in compliance is relatively small, we do not expect to see differences in the outcome. Nevertheless, we looked into the data, but again found nothing to report.

Missing Data Handling: Dropouts are described, but no imputation or sensitivity analysis is conducted.

This is now clearly stated. Dropouts’ data were not used. L385f

While the study presents a novel and potentially promising approach to tinnitus treatment, its methodological and statistical limitations significantly weaken the strength of its conclusions. The findings should be considered preliminary, and further research with more rigorous design and analysis is essential before LINTS can be recommended as a clinical intervention.

We stated this now in the Conclusion section (L808ff) and encourage independent studies with LINTS (L755ff)

Reviewer 3 Report

Comments and Suggestions for Authors

This study explores Low Intensity Noise Tinnitus Suppression (LINTS) as a novel approach for tinnitus treatment in a randomized, placebo-controlled Phase II clinical trial. The research question is clinically relevant and grounded in a strong theoretical framework, but the manuscript requires major revision for clarity, transparency, and balance in interpretation.

The introduction is overly detailed and focuses too much on theoretical mechanisms rather than the clinical rationale for the trial. It should be shortened and rewritten to emphasize the study’s objectives, hypothesis, and specific gap in tinnitus management that LINTS aims to address.

The description of study procedures, including randomization and blinding, is difficult to follow. A clear flow diagram is needed, and the rationale for unequal group sizes should be provided, along with an explanation of how randomization controlled for age, tinnitus duration, or hearing loss differences.

The placebo condition seems easily distinguishable from the active treatment, which could compromise blinding. The manuscript should explicitly acknowledge this limitation and discuss how perceptual differences between stimuli might have influenced patient expectations or dropout rates.

Outcome measures are insufficiently justified. The authors should explain why the Tinnitus Health Questionnaire was chosen as the primary endpoint and what threshold of improvement constitutes clinical relevance. Reporting should include effect sizes and confidence intervals rather than relying solely on p-values.

Statistical reporting is overly complex, and many non-significant results are discussed in detail. The results should be condensed to highlight key findings, removing speculative interpretations and ensuring that multiple comparisons are properly corrected.

Interpretation of the findings tends to overstate efficacy, especially regarding “complete tinnitus suppression.” These statements should be moderated and supported by data, emphasizing that the effect was limited to a few individuals rather than a general outcome.

The use of different devices to deliver the sound stimuli introduces variability that needs to be addressed. The authors should analyze whether treatment effects differed depending on the device type and discuss how this variability might affect reproducibility.

Participant adherence and missing data handling are unclear. The manuscript should specify how usage time was monitored, how dropouts were treated in the analysis, and whether an intention-to-treat approach was applied.

The writing is overly technical and could be streamlined for readability. Simplify sentences, reduce jargon, and move excessive theoretical and statistical details to supplementary materials to improve accessibility for a clinical audience.

The conclusions should be reframed to avoid overstating efficacy and should instead highlight LINTS as a promising but preliminary intervention requiring independent replication and larger-scale validation before clinical application.

Include the GAMER statement checklist with reference from this study to disclose any GAI usage: https://ebm.bmj.com/content/early/2025/05/13/bmjebm-2025-113825

Overall, this is a promising and conceptually innovative study, but it requires major revisions to improve methodological clarity, strengthen statistical validity, and moderate interpretative claims.

Author Response

The introduction is overly detailed and focuses too much on theoretical mechanisms rather than the clinical rationale for the trial. It should be shortened and rewritten to emphasize the study’s objectives, hypothesis, and specific gap in tinnitus management that LINTS aims to address.

We added the specific hypotheses to the introduction in more detail (L105ff) but did not shorten it, as we think, the background is needed to understand the approach.

The description of study procedures, including randomization and blinding, is difficult to follow. A clear flow diagram is needed, and the rationale for unequal group sizes should be provided, along with an explanation of how randomization controlled for age, tinnitus duration, or hearing loss differences.

We added the flow-diagram to Figure 1. The randomization was done for the age, gender and tinnitus duration but we were not able to balance HL completely (cf. Results). All randomization was done via our in-house developed computer program. The unequal group size resulted from the problem of time of the study. To be honest, we basically ran out of time.

The placebo condition seems easily distinguishable from the active treatment, which could compromise blinding. The manuscript should explicitly acknowledge this limitation and discuss how perceptual differences between stimuli might have influenced patient expectations or dropout rates.

It is only distinguishable, if you know what to look for. The patients were told that the optimal noise, that has been determined for each individually might be inaudible or sound slightly different. Therefore, the patients did not suspect any other than the optimal stimulus. Only after the debriefing, they were told about the possible WN. And the dropouts in the PT group were of the WN as a reason, because they told us, it did not work and were frustrated. The dropouts of the TO group were of different reasons. This is already stated in the manuscript.

Outcome measures are insufficiently justified. The authors should explain why the Tinnitus Health Questionnaire was chosen as the primary endpoint and what threshold of improvement constitutes clinical relevance. Reporting should include effect sizes and confidence intervals rather than relying solely on p-values.

We added the justification in the Methods section. L131ff

Statistical reporting is overly complex, and many non-significant results are discussed in detail. The results should be condensed to highlight key findings, removing speculative interpretations and ensuring that multiple comparisons are properly corrected.

We refrained from reducing the statistics, as other reviewers ased for more detail. But we added the effect size and their interpretation to the tables, so clinical relevant medium and strong effects are easily visible.

Interpretation of the findings tends to overstate efficacy, especially regarding “complete tinnitus suppression.” These statements should be moderated and supported by data, emphasizing that the effect was limited to a few individuals rather than a general outcome.

We rephrased that claims and moderated them.

The use of different devices to deliver the sound stimuli introduces variability that needs to be addressed. The authors should analyze whether treatment effects differed depending on the device type and discuss how this variability might affect reproducibility.

We analyzed this already (cf. Figure 5), but we agree that the variance might be increased by this. We added this to the discussion. L701ff

Participant adherence and missing data handling are unclear. The manuscript should specify how usage time was monitored, how dropouts were treated in the analysis, and whether an intention-to-treat approach was applied.

We added the missing information to the manuscript. Missing data (dropouts) were discarded completely (L385f). Time usage was monitored by the logging of the HA themselves, this is already stated in the Methods and Results.

The writing is overly technical and could be streamlined for readability. Simplify sentences, reduce jargon, and move excessive theoretical and statistical details to supplementary materials to improve accessibility for a clinical audience.

We did not change the overall structure of the manuscript but tried to explain more about the underlying ideas. Especially the discussion is improved, so “also the clinical audience” should be able to understand the main results. We do not discuss the “minor” outcomes or the lack thereof.

The conclusions should be reframed to avoid overstating efficacy and should instead highlight LINTS as a promising but preliminary intervention requiring independent replication and larger-scale validation before clinical application.

We moderated the conclusion.

Include the GAMER statement checklist with reference from this study to disclose any GAI usage: https://ebm.bmj.com/content/early/2025/05/13/bmjebm-2025-113825

We added the statement of AI usage to the Authors contribution section.

Overall, this is a promising and conceptually innovative study, but it requires major revisions to improve methodological clarity, strengthen statistical validity, and moderate interpretative claims.

Thank you, we tried to moderate our claims.

Round 2

Reviewer 1 Report

Comments and Suggestions for Authors

The comments have been addressed. 

Reviewer 3 Report

Comments and Suggestions for Authors

The authors have improved the quality of the paper after addressing suggestions from the previous round of review. Therefore, I recommend acceptance.